# The Fabrication and Characterization of Pickering Emulsion Gels Stabilized by Sorghum Flour

**DOI:** 10.3390/foods11142056

**Published:** 2022-07-12

**Authors:** Linlin Song, Sheng Zhang, Benguo Liu

**Affiliations:** 1College of Life Science and Technology, Henan Institute of Science and Technology, Xinxiang 453003, China; cnusll@126.com; 2School of Food Science, Henan Institute of Science and Technology, Xinxiang 453003, China; 18738591585@163.com

**Keywords:** sorghum flour, Pickering emulsion, microstructure, mechanical properties, β-carotene protection

## Abstract

Pickering emulsion gels have potential application as solid fat substitutes and nutraceutical carriers in foods, but a safe and easily available food-derived particle emulsifier is the bottleneck that limits their practical application. In this study, the function of sorghum flour as a particle emulsifier to stabilize the oil-in-water (O/W) Pickering emulsion gels with medium chain triglycerides (MCT) in the oil phase was introduced. Sorghum flour had suitable size distribution (median diameter, 21.47 μm) and wettability (contact angle, 38°) and could reduce the interfacial tension between MCT and water. The oil phase volume fraction (*φ*) and the addition amount of sorghum flour (*c*) had significant effects on the formation of Pickering emulsion gels. When c ≥ 5%, Pickering emulsion gels with *φ* = 70% could be obtained. Microstructure analysis indicated that sorghum flour not only played an emulsifying role at the O/W interface but also prevented oil droplets from coalescing through its viscous effect in the aqueous phase. With increases in *c*, the droplet size of the emulsion gel decreased, its mechanical properties gradually strengthened, and its protective effect on β-carotene against UV irradiation also improved.

## 1. Introduction

Sorghum (*Sorghumbicolor (L.) Moench*) is an annual herb of the genus Gramineae sorghum. Because of its high yield, strong adaptability and resistance, it is widely planted in drought, waterlogged and saline areas all over the world [1,2]. It is the fifth-largest cereal crop in the world after wheat, corn, rice and barley [3]. It has a long history of cultivation around the world. Although sorghum has high nutritional value and good processing properties, its high tannin content, low protein digestibility and poor palatability have limited its in-depth development for food processing [2,3]. Currently, the majority of sorghum is used as a brewing ingredient or feed, and there are relatively few food applications, mostly through simple processing to produce traditional foods such as rice, steamed bread and porridge [4]. These traditional sorghum foods are generally in the primary processing stage, mostly made from sorghum after simple crushing. These sorghum products are coarse in taste and poor in quality, resulting in a lower acceptance of sorghum products. Therefore, the development of high-quality and diverse sorghum products is important to promote the expanded processing of sorghum.

Pickering emulsion gel is a gel-like emulsion stabilized by solid particles instead of surfactants [5]. Compared with traditional emulsion gel, Pickering emulsion gel also has some mechanical properties, but its production cost is lower. In addition, it is not easily affected by environmental factors (pH, salt ion concentration, temperature, etc.), and it has potential application in fat substitution and as a nutraceutical carrier [6,7]. At present, food-grade Pickering emulsion gels are mainly stabilized by colloidal particles prepared using biomacromolecules (proteins, polysaccharides, etc.) [8,9,10]. However, these colloidal particles have some shortcomings, such as a tedious preparation process and environmental pollution, so their application in the food field is limited [11,12]. The development of food-derived particle emulsifiers has become a hot spot in the field of food.

Cereal, an important source of human food, is a natural colloidal particle composed of polysaccharides, proteins and other components; it is also a potential Pickering emulsifier. Lu et al. [13,14] found that red rice flour and black rice flour could stabilize Pickering emulsion and significantly improve the antioxidant properties of the oil phase of the emulsion. Huang et al. [15] found that media-milled purple potato flour could also stabilize Pickering emulsions, and there was a synergistic effect between starch and fiber. In this study, the feasibility of stabilizing Pickering emulsion gels with sorghum flour was evaluated for the first time. The effect of sorghum’s addition amount on the mechanical properties and β-carotene protective effects of Pickering emulsion gels were also systematically investigated.

## 2. Materials and Methods

### 2.1. Materials and Chemicals

Sorghum was purchased from the local market, and medium-chain triglyceride (MCT) was the product of Yuanye Biotechnology Co., Ltd., (Shanghai, China). Ultrapure water was obtained from a Thermo Gen Pure ultraviolet (UV)/ultrafiltration water system (Waltham, MA, USA). All other chemicals were of analytical grade.

### 2.2. The Preparation and Chemical Composition Determination of Sorghum Flour

The sorghum was crushed through a 100-mesh sieve and collected. The water content, protein content, starch content, lipid content and mineral content of sorghum flour were determined according to AOAC 925.10, AOAC 992.23, AOAC 996.11, AOAC 923.05 and AOAC 923.03, respectively.

### 2.3. The Appearance Observation and Particle Size Determination of the Sorghum Flour

The sorghum flour was evenly distributed on the specimen stub with double adhesive tape and coated with a thin gold layer under vacuum using a Hitachi E-1010 ion sputter (Tokyo, Japan). Then, its appearance was observed at 2000× magnification with an FEI Quanta 200 environmental scanning electron microscope (Hillsboro, OH, USA). The particle size distribution of the sorghum flour was determined using a BETTER BT-9300H laser particle analyzer (Dandong, China).

### 2.4. The Determination of the Interfacial Tension of the Sorghum Flour

The effect of sorghum flour on the MCT/water interfacial tension was measured using a Theta Lite optical contact angle meter (Biolin Scientific, Stockholm, Sweden). The excess sorghum flour was added to ultra-pure water, ultrasonicated at room temperature for 10 min and then allowed to rest for 15 min. The obtained supernatant was loaded into the syringe. Then, a drop of supernatant solution was added to a cuvette containing MCT. The drop shape was recorded continuously and fitted with the Young–Laplace equation to calculate the interfacial tension. The solution with ultra-pure water instead of supernatant was also measured as control.

### 2.5. The Determination of the Contact Angle of the Sorghum Flour

The contact angle of the sorghum flour was measured with a Theta Lite optical contact angle meter (Biolin Scientific, Stockholm, Sweden). The sample was pressed into a film at the pressure of 20 MPa and placed on a specimen stub. A drop of ultra-pure water was released onto the film surface. Then, the drop shape was recorded continuously and fitted with the Ellipse equation to calculate the contact angle.

### 2.6. The Preparation of the Pickering Emulsion Gels

A certain amount of sorghum flour was dispersed in water as aqueous phase and MCT as oil phase. The two phases were mixed at room temperature and homogenized at 12,000 r/min for 3 min to prepare Pickering emulsions with *φ* = 50, 60, 70, 80, 90% and *c* = 1.0, 3.0, 5.0, 7.0, 9.0%. The formation of the emulsion gel was confirmed using the inverted-tube method.

### 2.7. The Microscopic Observation of the Pickering Emulsion Gels

A total of 20 μL of the emulsion gel developed at *φ* = 70%, *c* = 5, 7, 9% was deposited onto a slide and observed by a BH200P polarizing microscope (Shanghai Sunny Hengping Scientific Instrument Co., Ltd., Shanghai, China). The droplet size was analyzed in ImageJ software (National Institutes of Health, Bethesda, MD, USA) based on a published method [16].

### 2.8. The Measurement of the Gel Strength of the Pickering Emulsion Gels

The gel strength values of the Pickering emulsion gels developed at *φ* = 70% and *c* = 5, 7 and 9% were determined based on the GMIA gelation mode using a TA-XT Plus texture analyzer (Stable Micro Systems, Surrey, UK). The sample in the cylindrical glass bottle was penetrated with a P/0.5 probe. The trigger force was set at 2.0 g, and the distance was 10 mm. The pretest speed, test speed, and posttest speed were set to 1.5, 1.0 and 1.0 mm/s, respectively.

### 2.9. Measurement of Microrheological Properties of Pickering Emulsion Gels

The microrheological properties of the Pickering emulsion gels developed at *φ* = 70% and *c* = 5, 7 and 9% were measured using a Rheolaser LAB6 microrheometer (Formulation, France). The freshly prepared emulsion in the amount of 20 mL was added to the 25 mL test bottle. The test bottle was placed in the chamber of microrheometer and monitored by a charge-coupled device (CCD) at 25 °C for 6 h. The obtained data were processed in RheosoftMaster1.4.0.0 software to calculate elasticity index (EI) and macroscopic viscosity index (MVI).

### 2.10. Determination of β-Carotene Protective Capacity of Pickering Emulsion Gels

With the MCT containing 2.0 mg/mL β-carotene as the oil phase, the Pickering emulsion gels developed at *φ* = 70% and *c* = 5, 7 and 9% were prepared according to Section 2.7. The obtained sample was placed under a UV lamp (power, 6 W) at the distance of 10 cm in an incubator at 30 °C. Every 24 h, 1 mL of the sample was extracted and mixed with a 5-mL mixture of hexane and ethanol (2:1, *v*/*v*). The absorbance of the diluted solution at 450 nm was read. Then, its β-carotene content was determined by comparison with the β-carotene standard curve. The corresponding β-carotene retention rate could be obtained [17]. The MCT with the same β-carotene content was used as the control.

### 2.11. Statistical Analysis

The experimental results were expressed as mean ± standard deviation (*n* = 3). The Tukey method was used for significance analysis with confidence level of 95% using the SPSS 18 software package (SPSS Inc., Chicago, IL, USA).

## 3. Results and Discussion

### 3.1. The Characterization of the Sorghum Flour

As shown in Table 1, the chemical composition of sorghum flour was similar to that of grains such as wheat flour and rice. Its protein and starch contents were 8.50 ± 0.09% and 68.28 ± 0.80%, respectively, which were inferior to those of wheat flour. However, its lipid (2.91 ± 0.06%) and mineral content (1.93 ± 0.04%) were slightly higher than those of wheat flour [18].

The formation of Pickering emulsion is affected by many factors, such as the shapes, sizes, wettability and concentrations of solid particles, the pH, the ionic strength of the aqueous phase, the properties and volume fraction of the oil phase, etc. [19,20], in which the particle emulsifier is generally measured in nanometers or microns. Therefore, we investigated the appearance and size distribution of the sorghum flour particles. As shown in Figure 1, most of the particles were spherical and oval, and a few were irregular; the particle size was in the range of 0.3–75 μm, showing a unimodal pattern with the median diameter of 21.47 μm. It is reported that the droplet size of Pickering emulsions stabilized by starches is 1–100 μm [8], so the size of the sorghum flour meets the requirements, and it has the potential to stabilize Pickering emulsions.

### 3.2. The Wettability of the Sorghum Flour

The Pickering emulsifying ability of particles is closely related to the wettability of particles, which allows particles to aggregate spontaneously at the oil–water interface and maintain the aggregation through volume exclusion and spatial hindrance [21]. The wettability of particles can be evaluated by reducing the oil/water interfacial tension and contact angle [20]. As shown in Figure 2, compared with the control (24.97 ± 0.51 mN/m), the sorghum flour could significantly reduce the oil/water interfacial tension (19.54 ± 0.47 mN/m). Although the main component of the sorghum flour was starch, it still contained a certain amount of natural emulsifiers, such as lipids and proteins, which could lead to the reduction of the oil/water interfacial tension. Previous studies have also shown that starch has the ability to stabilize Pickering emulsions when combined with proteins or lipids [22,23]. The contact angle can reflect the wetting effect of solid particles in oil and water phases. For spherical particles, when 15° < θ < 90°, it can stabilize O/W Pickering emulsions [20]. In this study, the contact angle of the sorghum flour was 38° (Figure 3). Combined with the interfacial tension analysis, it indicated that the sorghum flour had certain wettability and could stabilize the fabrication of O/W Pickering emulsions.

### 3.3. The Formation of the Pickering Emulsion Gels

In order to evaluate the feasibility of fabricating Pickering emulsion gels with sorghum flour, the effects of *c* (1–9%) and *φ* (50–90%) on the formation of the emulsion gels were investigated. As shown in Figure 4, when *φ* = 50%, Pickering emulsion gels could not be obtained in the experimental *c* range; when *φ* = 60%; the gel could only be developed at *c* = 9%; when *φ* = 70%, Pickering emulsion gels could be formed at *c* = 5, 7 and 9%. When *φ* = 80 or 90%, the emulsion was stratified. For purpose of revealing the corresponding formation mechanism, the Pickering emulsion gels developed at *φ* = 70% and *c* = 5, 7 and 9% were selected for the follow-up experiments.

### 3.4. Microscopic Observation Analysis of Pickering Emulsion Gels

In this study, the prepared Pickering emulsion gels developed at *φ* = 70% and *c* = 5, 7 and 9% dispersed in water but not in MCT, indicating that they were O/W emulsions, which coincided with the wettability analysis. Figure 5 demonstrates the effect of *c* on the droplet sizes of the emulsion gels. With increasing *c*, the droplet size decreased significantly. There was linear negative correlation between the two (r^2^ = 0.8771). Yan et al. [24] found a similar phenomenon. This is because when *c* is low, the coverage of particle emulsifier on the surface of oil droplets is not enough to prevent the merging of oil droplets, so the oil droplets are larger. When the coverage of oil droplets increases with the further increase of *c*, the dense interfacial particle layer can form outside the droplets, thus inhibiting further merging, and the droplets are smaller.

### 3.5. The Gel Strength of the Pickering Emulsion Gels

The strength of emulsion gel not only reflects its structural compactness but also is closely related to its processing properties and application fields [16]. In this study, the effect of *c* on the gel strength was investigated (Figure 6). The gel strengths of Pickering emulsion gels developed at *c* = 5, 7 and 9% and *φ* = 70% were 3.71 ± 0.21 g, 5.42 ± 0.16 g and 6.83 ± 0.10 g, respectively. Increasing *c* leads to increasing of gel strength. There was a linear positive correlation between them (r^2^ = 0.9391). This is due to the fact that with the increase in *c*, the droplet size decreases gradually, and the smaller oil droplets are beneficial to the compact arrangement, which significantly improves the strength of the emulsion gel; there is a linear negative correlation between the oil droplet size and the gel strength (r^2^ = 0.9115). Therefore, the higher *c*, the more stable the emulsion gel.

### 3.6. The Microrheological Analysis of the Pickering Emulsion Gels

Microrheology is a rheological technique based on diffusion spectroscopy (DWS) that can reflect rheological behavior without mechanical shear [25]. In the process of measurement, the CCD probe tracks the Brownian motion of the particles by a coherent laser emitted into the emulsion. The obtained EI and MVI correspond to G′ and G″ in the traditional mechanical rheology, respectively, reflecting the elasticity and viscosity of the system [26,27]. Here, the EI and MVI of all samples increased gradually with time (Figure 7), which was due to the viscosity effect caused by starch. In addition, the EI and MVI of the samples were positively correlated with *c*, and the EI of the emulsion gel developed at *φ* = 70% and *c* = 5% fluctuated greatly, indicating that the gel was weak; this was consistent with the results of microscopic observation and the gel strength analysis. At 6 h, EI and MVI were positively correlated with the gel strength, with r^2^ of 0.9370 and 0.9469, respectively, indicating that the microrheological results and the gel strength were mutually confirmed.

### 3.7. The β-Carotene Protective Capacity of the Pickering Emulsion Gels

β-Carotene is a common vitamin A supplement with the function of improving vision, but its application is limited because of its unstable exposure to oxygen, heat and light [28]. In this study, the β-carotene protective capacity of Pickering emulsion gels stabilized by sorghum flour against UV irradiation was evaluated. Figure 8 displays the effect of *c* on the β-carotene retention rate. Compared with MCT, the Pickering emulsion gel stabilized with sorghum flour had obvious protective effects on loaded β-carotene. On the 3rd day after UV irradiation, the β-carotene retention rate in MCT was close to zero, while the β-carotene retention rate of the Pickering emulsion gel was still higher than 75%. On the 9th day, the β-carotene retention rates of the Pickering emulsion gel with *c* = 7% and 9% were similar and significantly higher than that of the Pickering emulsion gel with *c* = 5%, indicating that *c* was positively correlated with β-carotene retention. The mechanism of protecting β-carotene using Pickering emulsion gels stabilized with sorghum flour may be as follows: (1) The sorghum flour adsorbed on the O/W interface forms a shell structure that weakens the exposure of UV to β-carotene in MCT; (2) the sorghum flour contains a small amount of tannins [29,30], which have UV absorption and help to improve the protective effect of β-carotene.

## 4. Conclusions

In this study, sorghum flour had suitable size distribution and wettability and could stabilize MCT-based O/W Pickering emulsion gels. The microstructure analysis showed that the sorghum flour not only played an emulsifying role at the interface but also prevented oil droplets from coalescing through its viscous effect in the aqueous phase. With increasing sorghum flour concentration, the droplet size of the emulsion gel decreased, its mechanical properties (gel strength, EI and MVI) increased gradually, and the protective effect on β-carotene also increased. The obtained results not only provide a reference for the deep processing of sorghum but also promote the green and efficient preparation of food-grade Pickering emulsion gels.

## Figures and Tables

**Figure 1 foods-11-02056-f001:**
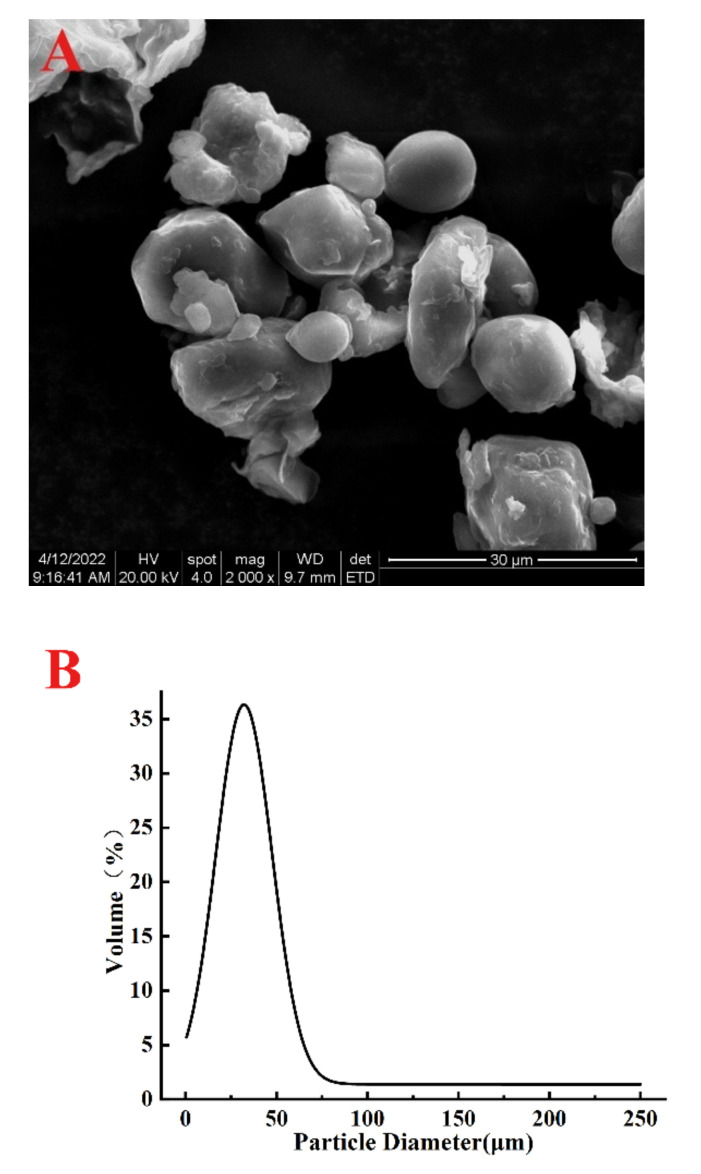
SEM image (**A**) and size distribution (**B**) of sorghum flour.

**Figure 2 foods-11-02056-f002:**
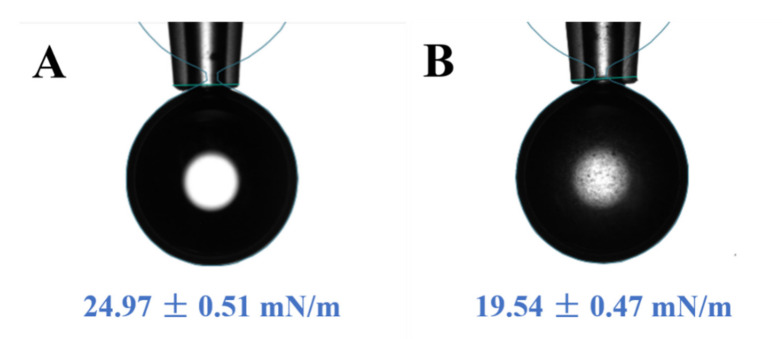
The effect of sorghum flour on the interfacial tension between MCT and water (**A**: control; **B**: sorghum flour).

**Figure 3 foods-11-02056-f003:**
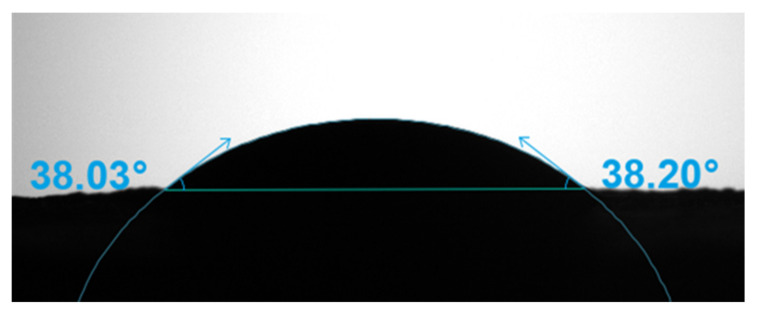
The water-in-air contact angle of the sorghum flour.

**Figure 4 foods-11-02056-f004:**
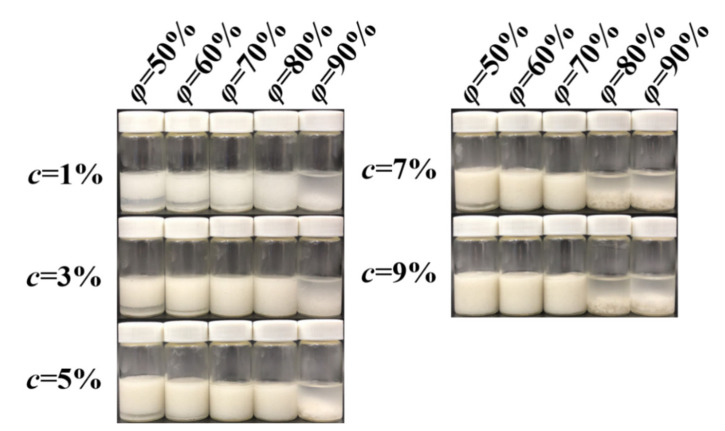
The formation of Pickering emulsions stabilized by sorghum flour.

**Figure 5 foods-11-02056-f005:**
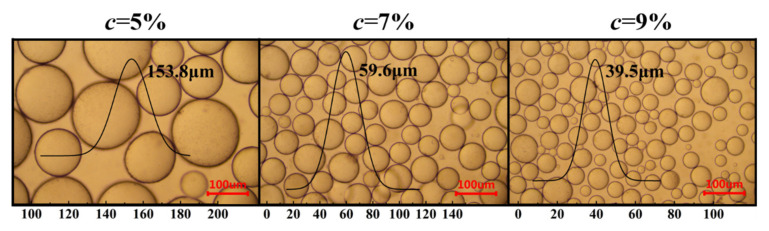
Microscopic observation of the Pickering emulsion gels developed by sorghum flour at *φ* = 70% and *c* = 5, 7 and 9%.

**Figure 6 foods-11-02056-f006:**
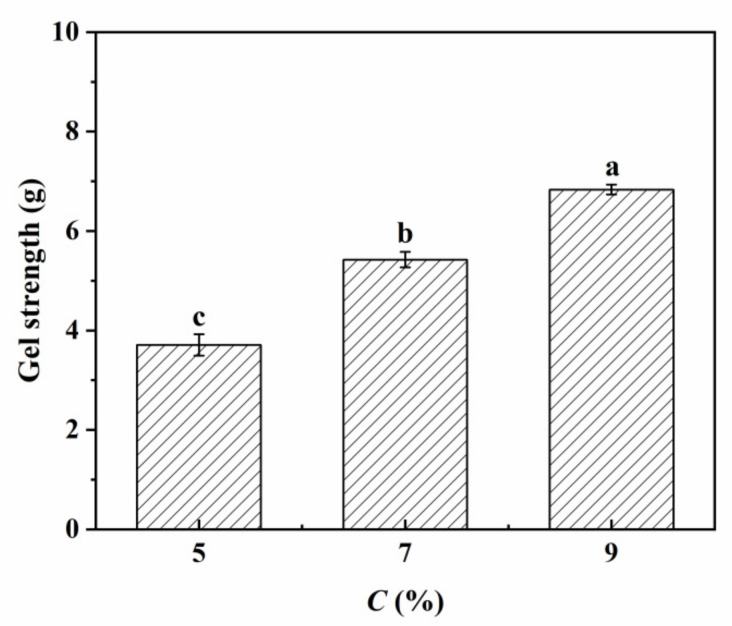
Gel strength values of the Pickering emulsion gels developed by sorghum flour at *φ* = 70% and *c* = 5, 7 and 9%.

**Figure 7 foods-11-02056-f007:**
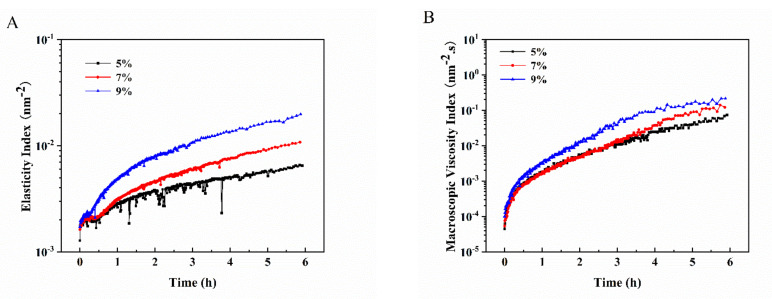
The EI (**A**) and MVI (**B**) of the Pickering emulsion gels developed by sorghum flour at *φ* = 70% and *c* = 5, 7 and 9%.

**Figure 8 foods-11-02056-f008:**
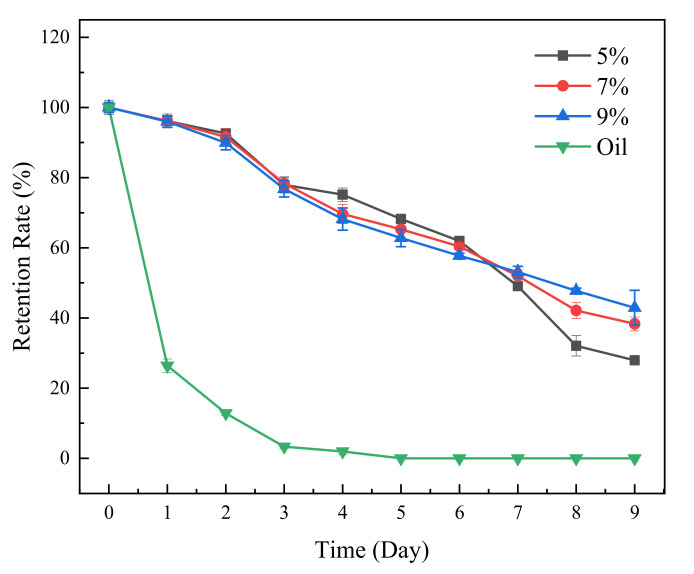
Light stabilities of β-carotene under UV irradiation in the oil and Pickering emulsion gels developed by sorghum flour at *φ* = 70% and *c* = 5, 7 and 9%.

**Table 1 foods-11-02056-t001:** The chemical composition of the sorghum flour on wet basis.

Water (%)	Protein (%)	Starch (%)	Lipid (%)	Mineral (%)
10.58 ± 0.08	8.50 ± 0.09	68.28 ± 0.80	2.91 ± 0.06	1.93 ± 0.04

## Data Availability

The data that support the findings of this study are available on request from the corresponding author. The data are not publicly available due to privacy or ethical restrictions.

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
