# Peer review of "The Fabrication and Characterization of Pickering Emulsion Gels Stabilized by Sorghum Flour"

_foods, 2022, doi:10.3390/foods11142056_

Round 1

Reviewer 1 Report

The manuscript of Linlin Song et al. describes the possible exploit of sorghum flour as a particle emulsifier to stabilize the oil-in-water Pickering emulsion gels with medium-chain triglycerides. The work description is unambiguous; the objectives are well defined, and the results are well exposed in a rigorously scientific manner. Moreover, the text is precise and never verbose or redundant. In addition, the topic is captivating and may be of great interest to readers. I found only a few concerns that forced me to ask for major revisions. The first is regarding the use (and the digestibility) of uncooked cereal flour as a food ingredient. The authors should consider this aspect by at least planning the simulated digestion of a product containing the sorghum stabilized oil-in-water emulsion. A second detail regards the fibre content of the used flour; why the Authors do not determine this parameter? In addition, the Authors shud furnish additional information about the sorghum used in terms of the variety and technological procedures used in its preparation (e.g. was it hulled?). If the germ should be present, the Authors must consider the potentially harmful effects of lipases on emulsified oils during storage. A further interesting aspect of the results of this work is the protective effect against the oxidation of β-carotene. Perhaps this observation deserves more emphasis and should be stated in the manuscript title or, at least, among the keywords. The Authors ascribe this effect to the tannins, which could absorb the UV. All sorghum varieties contain many phenolic acids, and most contain flavonoids. The total phenol content (in both flavonoids and phenolic acids) is correlated with general antioxidant activity. This antioxidant activity is different in the different sorghum varieties, and this is an additional reason to give more specifications on the raw material they used. The Authors, considering this effect, should add an, at least partial, characterization of the phenols profile and antioxidant activity of the flour they used.

Author Response

Response to Reviewer 1s Comments

The manuscript of Linlin Song et al. describes the possible exploit of sorghum flour as a particle emulsifier to stabilize the oil-in-water Pickering emulsion gels with medium-chain triglycerides. The work description is unambiguous; the objectives are well defined, and the results are well exposed in a rigorously scientific manner. Moreover, the text is precise and never verbose or redundant. In addition, the topic is captivating and may be of great interest to readers. I found only a few concerns that forced me to ask for major revisions.

Question 1: The first is regarding the use (and the digestibility) of uncooked cereal flour as a food ingredient. The authors should consider this aspect by at least planning the simulated digestion of a product containing the sorghum stabilized oil-in-water emulsion.

Response 1: Thank you for your suggestion! Our manuscript introduced the feasibility of fabricating Pickering emulsion gel with φ=70% using sorghum flour. The resulting emulsion gel is not used for direct consumption. We are using the obtained Pickering emulsion gel with vegetable oil as oil phase to partly replace the butter to prepare the cake. We think this is its most promising application area. So we don’t consider its simulated digestion. Thank you!

Question 2: A second detail regards the fibre content of the used flour; why the Authors do not determine this parameter? In addition, the Authors shud furnish additional information about the sorghum used in terms of the variety and technological procedures used in its preparation (e.g. was it hulled?). If the germ should be present, the Authors must consider the potentially harmful effects of lipases on emulsified oils during storage.

Response 2: Thank you for your suggestion! The sorghum used is a commercial product from local market. It is well shelled and germ removed, which is supported by the low lipid content and high starch content in Table 1. So we don't measure its fiber content and lipase. Thank you for your reminder, we will pay attention to this problem in the research about the Pickering emulsions stabilized by whole cereals. Thank you!

Question 3: A further interesting aspect of the results of this work is the protective effect against the oxidation of β-carotene. Perhaps this observation deserves more emphasis and should be stated in the manuscript title or, at least, among the keywords. The Authors ascribe this effect to the tannins, which could absorb the UV. All sorghum varieties contain many phenolic acids, and most contain flavonoids. The total phenol content (in both flavonoids and phenolic acids) is correlated with general antioxidant activity. This antioxidant activity is different in the different sorghum varieties, and this is an additional reason to give more specifications on the raw material they used. The Authors, considering this effect, should add an, at least partial, characterization of the phenols profile and antioxidant activity of the flour they used.

Response 3: Thank you for your suggestion! The main objective is to evaluate the feasibility of fabricating Pickering emulsion gel with sorghum flour. Like the gel strength test, β-carotene protection test is also a means to reflect the gel structure. So we don’t emphasize it in the title. But according to your suggestion, we add “β-carotene protection” in keywords.

At present, there are many studies on sorghum polyphenols, which are closely related to the biological activities of sorghum, but this study is mainly focused on fabrication and characterization of Pickering emulsion gels stabilized by sorghum flour. Therefore, sorghum polyphenols were not investigated in this study. Thank you for our understanding.

Reviewer 2 Report

Fabrication and Characterization of Pickering Emulsion Gels  Stabilized by Sorghum Flour is interesting but some parts need to be improved.

-The general remark: results should be described and decided in more detailed way.

-There many type of measurement but not too many data (low number of analysed variants).

Authors should compare all properties and find some correlation between them.

-Conclusion are more statements.

-Chapter 2.8. What kind o mechanical test was used? (compression, penetration). Was the sample in container. What level of deformation of gel was used?

Author Response

Response to Reviewer 2’s Comments

Fabrication and Characterization of Pickering Emulsion Gels  Stabilized by Sorghum Flour is interesting but some parts need to be improved.

Question 1: The general remark: results should be described and decided in more detailed way.

Response 1: Thank you for your suggestion! We strengthened the “results and discussion” section. These revisions are marked in red. Thank you!

Question 2: There many type of measurement but not too many data (low number of analysed variants).

Response 2: Thank you for your suggestion! We have strengthened the discussion with data. Thank you!

Question 3: Authors should compare all properties and find some correlation between them.

Response 3: Thank you for your suggestion! We have strengthened the discussion of the correlation between various kinds of properties. Thank you!

Question 4: Conclusion are more statements.

Response 4: Thank you for your suggestion! We have strengthened the conclusion section. Thank you!

Question 5: Chapter 2.8. What kind o mechanical test was used? (compression, penetration). Was the sample in container. What level of deformation of gel was used?

Response 5: Thank you for your suggestion! We have revised section 2.8. The more experimental descriptions have been added. Thank you!

Reviewer 3 Report

The research undertaken by the authors is interesting and provide some new aspects of utilising sorghum flour in food technology. The scope of the research is sufficient; however, the discussion of the results could be somewhat extended (see comments below). Moreover, I would recommend providing more information on mechanism of forming Pickering emulsion in the Introduction section as well as explain the difference, if exists, between emulsion and emulsion gel. Additional remarks are provided below.

L53: Should be: Liu et al. [number]

L55: Should be: Huang et al. [number]

L70-71: The references (Chinese Standards) are not universal ones and may not be available to the reader. It is recommended to replace them with AOAC standards.

Section 2.8: Please provide details of the measurement. Was the sample in any container? Was the sample penetrated or compressed? What was the percentage of the sample compression/penetration?

L116: Expand the CCD abbreviation.

Section 2.10: Please provide the principle of the [17] method.

Table 1: Please specify if the amounts of the determined compounds are expressed per dry or wet mass of the sorghum flour.

L178: Should not there be also c = 5%?

L107-108, 113-114, 121: There is inconsistency with the information provided in L180-181.

L189: Put the reference just after the author’s surname.

L192: Replace dot (after c) with comma.

Figure 6: Please add the meaning of the bars and mark the columns with letters indicating the statistical differences between them.

L215-216: Please explain a mechanism of the viscosity effect of the sorghum starch in time.

L228: Write the c in italic.

L244: Was suitable for what?

L249: What does “mechanical properties increased” mean? Values of specified parameters can increase.

Conclusions: Please provide any examples of food that can be prepared with such Pickering emulsion gels.

Author Response

Response to Reviewer 3’s Comments

Question 1: The research undertaken by the authors is interesting and provide some new aspects of utilising sorghum flour in food technology. The scope of the research is sufficient; however, the discussion of the results could be somewhat extended (see comments below).

Response 1: Thank you for your suggestion! We have revised this manuscript according to you requirements. These revisions are marked in red. Thank you!

Question 2: Moreover, I would recommend providing more information on mechanism of forming Pickering emulsion in the Introduction section as well as explain the difference, if exists, between emulsion and emulsion gel. Additional remarks are provided below.

Response 2: Thank you for your suggestion! We have provided more information on mechanism of forming Pickering emulsion in the introduction section. Thank you!

Question 3: L53: Should be: Liu et al. [number]

Response 3: Thank you for your suggestion! We have revised it, “Lu et al. [13-14] ”. Thank you!

Question 4: L55: Should be: Huang et al. [number]

Response 4: Thank you for your suggestion! We have revised it, “Huang et al. [15]”. Thank you!

Question 5: L70-71: The references (Chinese Standards) are not universal ones and may not be available to the reader. It is recommended to replace them with AOAC standards.

Response 5: Thank you for your suggestion! We have replaced these Chinese Standards with AOAC standards in this revised manuscript. Thank you!

Question 6: Section 2.8: Please provide details of the measurement. Was the sample in any container? Was the sample penetrated or compressed? What was the percentage of the sample compression/penetration?

Response 6: Thank you for your suggestion! We have revised the section 2.8. The more experimental descriptions have been added. Thank you!

Question 7: L116: Expand the CCD abbreviation.

Response 7: Thank you for your suggestion! We have expand the CCD abbreviation “a charge-coupled device”, in this revised manuscript. Thank you!

Question 8: Section 2.10: Please provide the principle of the [17] method.

Response 8: Thank you for your suggestion! We have revised section 2.10. The more experimental descriptions have been added. Thank you!

Question 9: Table 1: Please specify if the amounts of the determined compounds are expressed per dry or wet mass of the sorghum flour.

Response 9: Thank you for your suggestion! We revised the caption of Table 1 to specify that the measurements are expressed based wet basis. Thank you!

Question 10: L178: Should not there be also c = 5%?

Response 10: We are sorry for this mistake. We have added it! Thank you!

Question 11: L107-108, 113-114, 121: There is inconsistency with the information provided in L180-181.

Response 11: We are sorry for this mistake. We have revised it! Thank you!

Question 12: L189: Put the reference just after the author’s surname.

Response 12: Thank you for your suggestion! We have revised it, “Yan et al. [24] ”. Thank you!

Question 13: L192: Replace dot (after c) with comma.

Response 13: Thank you for your suggestion! We have revised it, “c,”. Thank you!

Question 14: Figure 6: Please add the meaning of the bars and mark the columns with letters indicating the statistical differences between them.

Response 14: Thank you for your suggestion! We have marked the columns in Figure 6 with letters indicating the statistical differences between them. Thank you!

Question 15: L215-216: Please explain a mechanism of the viscosity effect of the sorghum starch in time.

Response 15: Thank you for your question! This is because with the extension of time, the starch particles fully absorb water, and the starch molecules are continuously released into the water, which improves the viscosity of the aqueous solution. Thank  you!

Question 16: L228: Write the c in italic.

Response 16: Thank you for your suggestion! We have revised it. Thank you!

Question 17: L244: Was suitable for what?

Response 17: Thank you for your suggestion! We have revised this expression! Thank you!

Question 18: L249: What does “mechanical properties increased” mean? Values of specified parameters can increase.

Response 18: Thank you for your suggestion! The mechanical properties included gel strength, EI and MVI. We have revised this expression. Thank you!

Question 19: Conclusions: Please provide any examples of food that can be prepared with such Pickering emulsion gels.

Response 19: Thank you for your question! We are using the obtained Pickering emulsion gel with vegetable oil as oil phase to partly replace the butter to prepare the cake. We think this is its most promising application area. Thank you!

Round 2

Reviewer 1 Report

The manuscript of Linlin Song et al., in its revised form, is now looks good.

Author Response

Thank you for your positive comment!

Reviewer 2 Report

The article was not improved in sufficient way. The chapter discussion still needs expansion. Also correlation between different properties (wettability, mechanical parameters, average droplet size)  should be analysed.

Author Response

Fabrication and Characterization of Pickering Emulsion Gels  Stabilized by Sorghum Flour is interesting but some parts need to be improved.

Question 1: The article was not improved in sufficient way. The chapter discussion still needs expansion. Also correlation between different properties (wettability, mechanical parameters, average droplet size)  should be analysed.

Response 1: Thank you for your suggestion! We have added the corresponding discussion, and the correlation between different properties. These revisions are marked in red. Thank you!